# Effect of Methionine Restriction on Aging: Its Relationship to Oxidative Stress

**DOI:** 10.3390/biomedicines9020130

**Published:** 2021-01-29

**Authors:** Munehiro Kitada, Yoshio Ogura, Itaru Monno, Jing Xu, Daisuke Koya

**Affiliations:** 1Department of Diabetology and Endocrinology, Kanazawa Medical University, Uchinada, Ishikawa 920-0293, Japan; namu1192@kanazawa-med.ac.jp (Y.O.); imonno@kanazawa-med.ac.jp (I.M.); xujing@kanazawa-med.ac.jp (J.X.); 2Division of Anticipatory Molecular Food Science and Technology, Medical Research Institute, Kanazawa Medical University, Uchinada, Ishikawa 920-0293, Japan

**Keywords:** methionine restriction, oxidative stress, autophagy, lifespan extension, metabolic health

## Abstract

Enhanced oxidative stress is closely related to aging and impaired metabolic health and is influenced by diet-derived nutrients and energy. Recent studies have shown that methionine restriction (MetR) is related to longevity and metabolic health in organisms from yeast to rodents. The effect of MetR on lifespan extension and metabolic health is mediated partially through a reduction in oxidative stress. Methionine metabolism is involved in the supply of methyl donors such as S-adenosyl-methionine (SAM), glutathione synthesis and polyamine metabolism. SAM, a methionine metabolite, activates mechanistic target of rapamycin complex 1 and suppresses autophagy; therefore, MetR can induce autophagy. In the process of glutathione synthesis in methionine metabolism, hydrogen sulfide (H_2_S) is produced through cystathionine-β-synthase and cystathionine-γ-lyase; however, MetR can induce increased H_2_S production through this pathway. Similarly, MetR can increase the production of polyamines such as spermidine, which are involved in autophagy. In addition, MetR decreases oxidative stress by inhibiting reactive oxygen species production in mitochondria. Thus, MetR can attenuate oxidative stress through multiple mechanisms, consequently associating with lifespan extension and metabolic health. In this review, we summarize the current understanding of the effects of MetR on lifespan extension and metabolic health, focusing on the reduction in oxidative stress.

## 1. Introduction

Aging is a universal process that affects all organs and decreases their function. Age-related cellular dysfunction leads to failure to maintain cellular homeostasis, resulting in a decline in the responsiveness to intracellular stresses, including oxidative stress. Enhanced oxidative stress is implicated in the pathogenesis of the aging process, age-related disease and the impairment of metabolic health [1]. Therefore, the attenuation of oxidative stress is a therapeutic target against aging, age-related disease and metabolic health.

All organisms adapt to the alteration of nutrients that are available in the environment. A network of nutrients and nutrient-sensing pathways regulates cellular activities, such as metabolism, growth and aging [2]. As a dietary intervention, calorie restriction (CR) without malnutrition has been recognized as one of the experimental methods for lifespan extension and the improvement of metabolic health in organisms, partially through a reduction in cellular oxidative stress [3]. However, recent studies indicate that the restriction of proteins in the diet, not CR, can increase lifespan and suppress the incidence of age-related disease [4,5]. Among the restriction of a certain amino acid, methionine restriction (MetR) is thought to exert its benefits on lifespan extension and metabolic health [6,7,8,9]. Therefore, the benefit of protein restriction on lifespan extension and metabolic health may be exerted through MetR. In addition, in yeast, glucose restriction down-regulates the transcription and translation of methionine biosynthetic enzymes and transporters, leading to a decreased intracellular methionine concentration and lifespan extension [10]. Oxidative stress is closely related to age and impaired metabolic health. Therefore, MetR-induced beneficial effects on lifespan extension and metabolic health are mediated partially through a reduction in oxidative stress via multiple mechanisms [11], as described as follows: (1) MetR leads to the induction of autophagy due to the suppression of mechanistic target of rapamycin complex 1 (mTORC1) and increased polyamine; (2) MetR also decreases reactive oxygen species (ROS) production in mitochondria; and (3) MetR increases the production of hydrogen sulfide (H_2_S), a molecule that is indispensable for lifespan extension through CR. These mechanisms are induced by MetR and contribute to a reduction in oxidative stress, which is associated with lifespan extension and metabolic health. In this review, we summarize the current understanding of the effects of MetR on lifespan extension and metabolic health, focusing on the reduction in oxidative stress.

## 2. Metabolism of Methionine

Methionine is an essential amino acid that is necessary for normal growth and development, and functions as an initiator of protein synthesis. Methionine is also a sulfur-containing amino acid (SAA). In humans, methionine is obtained from both food and gut microbes, and is also supplied by autophagy, as described herein. In addition, metabolites of methionine contribute to many metabolic processes, including the synthesis of polyamine and nucleotide and glutathione production. Methionine metabolism-related pathways consist of three parts, namely, the methionine cycle pathway, the transsulfuration pathway and the salvage pathway [12].

### 2.1. Methionine Cycle Pathway

Methionine is catabolized by methionine adenosyltransferase 2A (MAT2A), generating S-adenosyl-methionine (SAM), which is a universal methyl donor. Methyltransferases (MTs) catalyze a variety of methylation reactions by the transfer of methyl groups on both histone proteins and nonhistone proteins [12] (Figure 1). On the other hand, glycine N-methyltransferase (Gnmt) produces S-adenosyl-homocysteine (SAH) from SAM (Figure 1). SAH hydrolase (SAHH, also known as adenosylhomocysteinase, AHCY) catalyzes the reversible hydrolysis of SAH to adenosine and l-homocysteine (Figure 1). Homocysteine is converted back to methionine by methionine synthase (MS), which requires 5-methyltetrahydrofolate as a methyl donor, or betaine homocysteine methyltransferase (BHMT), which requires betaine as a methyl donor, completing the methionine cycle [12] (Figure 1). In addition, homocysteine is converted to cysteine via the transsulfuration pathway.

### 2.2. Transsulfuration Pathway

Homocysteine produced through the methionine cycle is metabolized in the transsulfuration pathway to generate cysteine (Figure 1). Cystathionine-β-synthase (CBS) is the first-acting and rate-limiting enzyme of the transsulfuration pathway, which is the primary pathway for cysteine synthesis (Figure 1). CBS synthesizes cystathionine by the condensation of homocysteine and serine. Thereafter, cystathionine is hydrolyzed by cystathionine-γ-lyase (CGL) to generate cysteine, and cysteine is further utilized in the synthesis of glutathione (GSH) and taurine (Figure 1). In addition, homocysteine is catalyzed by CGL, producing homoserine and H_2_S, and both CGL and CBS catalyze H_2_S production from cysteine, as well as the production of pyruvate and serine, respectively (Figure 1). Thus, activation of the transsulfuration pathway can promote the production of H_2_S [12,13].

### 2.3. Salvage Pathway and Polyamine Biosynthesis

The methionine salvage pathway, which is also called the 5′-methylthioadenosine (MTA) cycle, is involved in the regeneration of methionine from SAM and the production of polyamines, such as spermidine [12,14] (Figure 1). In this pathway, SAM is decarboxylated by SAM decarboxylase 1 (AMD1) to decarboxylated SAM (dcSAM), which acts as an aminopropyl group donor. In parallel, arginine is converted by arginase (ARG) to ornithine and then decarboxylated by ornithine decarboxylase (ODC) to produce putrescine (Figure 1). Furthermore, putrescine is converted to spermidine and spermine through spermidine synthase (SRM) and spermine synthase (SMS), which use dcSAM as an aminopropyl donor, respectively (Figure 1). On the other hand, dcSAM is converted to MTA after the donation of an aminopropyl group for polyamine synthesis, and MTA is converted by multiple enzymatic steps back to methionine [12] (Figure 1).

## 3. Effects of MetR on Lifespan Extension and Metabolic Health

### 3.1. From Animal Studies

Dietary MetR has been demonstrated to extend the lifespan in yeast, *Drosophila*, *Caenorhabditis elegans*, mice and rats [6,7,8,9,15,16,17,18,19]. In addition to MetR, genetic MetR by knockdown of methionine synthase confers stress resistance in cultured fibroblasts, as well as a reduced doubling time and a replicative lifespan extension [20]. Orentreich et al. first reported that a reduction in the concentration of methionine from 0.86 to 0.17% in the diet (80% restriction of methionine) results in a 30% longer lifespan of Fischer 344 rats, although food intake was increased in the MetR-fed rats [6]. Similarly, Richie et al. also demonstrated that an 80% restriction of methionine resulted in a 44% increase in maximum lifespan and 43% lower body weight in Fischer 344 rats, compared to the controls [7]. In addition, feeding BALB/cJ × C57BL/6J F1 mice 65% MetR (a control diet of 0.43% methionine and a MetR diet of 0.15% methionine) led to lifespan extension [8]. MetR mice develop lens turbidity significantly slower and show age-related changes in T-cell subsets and lower serum insulin, glucose, insulin like growth factor-1 (IGF-1) and thyroxine (T4) levels [8]. Bárcena et al. showed that MetR prolongs the health span and longevity of 2 short-lived strains of mice with the *Lmna^G609G/G609G^* or *zmpste24*^−/−^ genotype, animal models of Hutchinson–Gilford progeria syndrome (HGPS), by ameliorating inflammation and DNA damage and restoring the lipid and bile acid levels [19].

Metabolically, MetR reduces adiposity in rodents, whereas, interestingly, both energy intake and energy expenditure (EE) are increased [21,22,23]. The increase in EE is induced by a process that compensates for increased energy intake, effectively limiting fat deposition. Furthermore, MetR increases metabolic flexibility and overall insulin sensitivity, and it improves lipid metabolism by decreasing systemic inflammation [21,22,23,24]. Malloy et al. showed that MetR (80%) decreases visceral fat and improves insulin sensitivity in aging Fischer 344 rats independent of energy restriction [21]. In addition, MetR reduces the leptin levels and increases the adiponectin levels in the blood of rats [21]. Ables et al. also demonstrated that MetR (control diet of 0.86% methionine and MetR diet of 0.12% methionine) protects high-fat diet (HFD)-fed mice against the development of obesity, insulin resistance and type 2 diabetes [24]. The effect of MetR is associated with a reduction in hepatic lipid accumulation and is enhanced in insulin sensitivity. On the other hand, Stone et al. showed that MetR (a control diet of 0.86% methionine and a MetR diet of 0.17% methionine) enhances hepatic insulin signaling in mice, whereas fibroblast growth factor 21 (FGF21) induced by MetR expressed in the liver enhances insulin signaling in adipocytes [25]. Lee et al. also reported that MetR (a control diet of 0.86% methionine and a MetR diet of 0.17% methionine) in 12-month-old mice completely reversed age-induced alterations in body weight, adiposity, physical activity, and glucose tolerance to the levels of 2-month-old control-fed mice, possibly through increased FGF21 [26]. Interestingly, Wang et al. demonstrated that MetR can prevent HFD-induced obesity and its associated metabolic disorders, such as impairment of the diurnal metabolism of lipids and bile acids, through the enhancement of the circadian rhythm in the liver and adipose tissue and maintenance of energy metabolism homeostasis in mice [27]. Moreover, MetR enhanced the diurnal expression of FGF21 and inhibited the accumulation of lipids, thereby leading to time-specific improvement in insulin resistance [27]. Regarding the amount of MetR, Forney et al. reported that restricting methionine from 0.17 to 0.25% is most effective in increasing EE, limiting fat deposition, reducing hepatic triglyceride synthesis and improving insulin sensitivity [28]. On the other hand, a diet providing 0.12% methionine is no more effective than a 0.17% methionine diet in improving these metabolic markers and may exhibit negative effects on body weight (BW) and lean mass compared to a 0.17% methionine diet. Thus, MetR is associated with metabolic benefits, including limiting the accumulation of fat, suppressing HFD-induced obesity, enhancing insulin sensitivity and preventing the development of diabetes.

### 3.2. From Human Studies

Plaisance et al. investigated the effects of MetR (2 mg methionine/kg BW/day) on the EE, body composition and metabolism in obese adults with metabolic syndrome for 16 weeks and compared the results to those obtained from adults on a control diet (33 mg methionine/kg BW/day) [29]. MetR improved insulin sensitivity and did not change the EE in either group; however, MetR led to a significant increase in fat oxidation and a decrease in intrahepatic lipid content [29]. In addition, Virtanen et al. showed that the relative risk for an acute coronary event in subjects with a high intake of methionine (>2.2 g methionine/day) was greater than that it was in people with a low intake of methionine (<1.7 g methionine/day) in a prospective cohort study (1981 men, aged 42–60 years at baseline, average 14-year follow-up) [30].

Furthermore, the SAM levels in plasma were related to higher fasting insulin levels, homeostasis model assessment of insulin resistance (HOMA-IR) and tumor necrosis factor-α (TNF-α) in a cross-sectional study involving 118 individuals with metabolic syndrome [31]. One report showed that the plasma SAM levels, but not methionine, were related to fat mass and truncal adiposity in a cross-sectional study involving 610 elderly subjects [32], and another report demonstrated that overfeeding raised the serum SAM in proportion to the fat mass gained in normal-to-overweight subjects [33]. Thus, increased SAM in blood associated with excess nutrition and accumulation of fat mass may be involved in whole-body metabolic impairment.

## 4. Mechanisms Underlying the Roles of MetR in Lifespan Extension and Metabolic Health Focusing on Antioxidative Stress

Numerous findings obtained from studies on MetR in the diet have revealed that methionine is associated with the regulation of the aging process and metabolism through multiple physiological and molecular mechanisms. The beneficial effect of MetR on lifespan extension and metabolic health is mediated partially through a reduction in oxidative stress via the induction of autophagy and H_2_S production and a reduction in free radical leakage from mitochondria.

### 4.1. Induction of Autophagy

Mitochondria are major sources of ROS, and mitochondria are attacked by ROS. Damaged mitochondria produce additional ROS, which leads to mitochondrial dysfunction and cellular aging. Autophagy is a lysosomal degradation pathway and plays an important role in the removal of damaged organelles, including mitochondria, to maintain cellular homeostasis [34]. However, impairment of autophagy is involved in the accumulation of damaged mitochondria in the cell, which results in enhanced overproduction of ROS in the mitochondria. Among the autophagy, damaged mitochondria are eliminated by the selective autophagy, mitophagy, leading to the induction of mitochondrial biogenesis and suppression of oxidative stress. Indeed, Plummer et al. reported that MetR-mediated chronological lifespan extension of a yeast requires the autophagic recycling of mitochondria; that is, mitophagy [35]. Therefore, adequate induction of autophagy can protect the cells against intracellular stresses associated with aging, such as oxidative stress, thereby leading to lifespan extension, i.e., longevity.

Autophagy is regulated by nutrient-sensing molecules, including mTORC1, a serine/threonine kinase subunit of mTOR that functions as a central regulator of cell growth and metabolism in response to changes in nutrient status [36]. mTORC1 is activated by growth factors, such as insulin, and amino acids, including methionine [36,37]. Activated mTORC1 phosphorylates the components of Ulk1 (Ser757), thereby inhibiting its activity and suppressing autophagy [36]. Recent reports show that methionine can activate mTORC1 via multiple pathways involved in the suppression of autophagy.

#### 4.1.1. Regulation of mTORC1 and Autophagy by Methionine

##### mTORC1 Activation through SAM Sensors

SAM is the metabolite of methionine that is converted by MAT2A as described above. Previous studies have demonstrated that SAM, not methionine, may be the main contributor to lifespan extension induced by MetR. Obata et al. showed that the enhancement of SAM catabolism by Gnmt activation exhibits *Drosophila* lifespan extension [38]. The levels of SAM are higher in old flies than in young flies, even though the expression of Gnmt is induced in a fork head box O (FOXO)-dependent manner in old flies [38]. However, Gnmt overexpression suppresses the age-dependent increase in SAM and extends the lifespan of old flies.

Previously, Gu et al. identified the mechanism by which methionine activates mTORC1 [39]. mTORC1 senses intracellular SAM (not methionine) levels, which triggers its activation (Figure 2A). SAMTOR is identified as a sensor for SAM, and SAMTOR binds with KICTOR, which is attached to lysosomes. SAM binds to SAMTOR, and SAMTOR dissociates from KICTOR. At the same time, GASTOR1 dissociates from SAMTOR, resulting in GASTOR1 inactivation. Since the GATOR1 complex has a RagA-binding ability and GTPase-activating protein (GAP) activity leading to RagA, the inactivation of GATOR1 leads to mTORC1 activation through an increase in RagA-GTP binding via GTPase inactivation. In contrast, the activation of GASTOR1 leads to the suppression of mTORC1 through the increase in RagA-GDP binding via its GTPase activation. Therefore, GASTOR1, which binds to SAMTOR and KICTOR, acts as a negative regulator of mTORC1 activation [36,37,39].

##### mTORC1 Activation through PP2A Activation

Another mechanism by which SAM induces mTORC1 activation has also been identified (Figure 2A). Sutter et al. demonstrated that SAM activates mTORC1 and suppresses autophagy through the methylation of phosphatase 2A (PP2A) [40]. In the presence of high levels of intracellular SAM, Ppm1 promotes the methylation of the PP2A catalytic subunit in response to intracellular SAM concentrations, which results in PP2A activation in yeast. In mammalian cells, the methylation of PP2A is catalyzed through a specific SAM methyltransferase, leucine carboxyl methyltransferase 1 (LCMT1) [41]. In yeast, methylated PP2A can inhibit Npr2 via dephosphorylation, leading to the activation of mTORC1 and the suppression of autophagy. SECIT in yeast consists of Npr2, Npr3 and Iml1 [42], and SECIT functions as a negative regulator of mTORC1 via GAP activation toward the yeast Rag orthologs Gtr1/2, which are members of the Rags family in mammals [43]. Therefore, in yeast, suppression of Seh1-associated complex inhibiting TORC1 (SEACIT) due to Npr2 dephosphorylation induced by PP2A activation results in mTORC1 activation. Similar to SEACIT in yeast, GASTOR1 in mammals consists of NPRL2, NPRL3 and DEPDC5; however, it remains unclear whether SAM-induced methylated PP2A is involved in mTORC1 activation through suppression of GASTOR1 via the inactivation of NPRL2 induced by its dephosphorylation, as observed for Npr2 in yeast. In contrast, at low intracellular SAM levels, the methylation levels of PP2A are decreased, thereby leading to the phosphorylation of Npr2 and consequently the suppression of mTORC1 activity and the induction of autophagy. Activation of PP2A possibly promotes the dephosphorylation of NPRL2 and results in the activation of mTORC1. However, it remains unclear whether PP2A is directly associated with the regulation of NPRL2 phosphorylation. Previously, we showed that a low-protein diet (LPD) improves diabetes-induced kidney injury; however, the addition of methionine to an LPD abrogated the benefits of the LPD in diabetic kidneys [44]. Specifically, LPD attenuated the increased levels of LCMT, methyl-PP2A and renal SAM in the kidneys of diabetic rats; however, the addition of methionine to the LPD increased the expression levels of LCMT, methyl-PP2 and SAM. In addition, the renal expression of Gnmt, a SAM-converting enzyme, in diabetic rats was lower than that in nondiabetic rats, but its expression levels did not change in the three groups: standard diet-fed, LPD-fed and LP + methionine diet-fed diabetic rats. Therefore, changes in renal SAM levels in diabetic rats were dependent on the content of dietary methionine [44]. Consistent with the changes in LCMT1 and methyl-PP2A, the activation of mTORC1 and suppression of autophagy were observed in standard diet-fed and LP + methionine-fed diabetic rats. Moreover, we confirmed that the methylation PP2A by SAM administration increased the level of methyl-PP2A and the activation of mTORC1 in cultured human kidney-2 (HK-2) cells [44].

##### Methionine Activates mTORC1 through TAS1R1/TAS1R3

Previously, a mammalian amino-acid taste receptor, the taste 1 receptor member 1 (TAS1R1)/Taste 1 receptor member 3 (TAS1R3) heterodimer, which is a cell-surface G protein-coupled receptor, was identified [45]. The receptor broadly senses most of the 20 amino acids. When sensing amino acids, the receptor activates mTORC1 through phospholipase C activation, an increase in intracellular calcium and mitogen-activated protein kinase 1/mitogen-activated protein kinase 3 activation [46]. TAS1R1–TAS1R3 is required for the mTORC1 localization to lysosomes as induced by amino acids; this lysosome localization of mTORC1 is required for its activation (Figure 2B). Several previous reports have demonstrated that TAS1R1–TAS1R3 can serve as a sensor for extracellular methionine in cultured C2C12 cells and bovine epithelial cells [47,48].

#### 4.1.2. Polyamines and Autophagy

Spermidine is a ubiquitous polyamine that is synthesized from putrescine and serves as a precursor of spermine, which stimulates cytoprotective autophagy [14]. External supplementation with spermidine extends the lifespan across species, from yeast, *C. elegans*, flies to mice [14]. In addition, supplementation with spermidine exerts cardioprotective effects, leading to a reduction in cardiac hypertrophy and the preservation of diastolic function in old mice through the induction of autophagy [49]. Spermidine promotes autophagy by suppressing several acetyltransferases, including EP300, one of the main negative regulators of autophagy, via the acetylation of autophagy-related genes (ATGs), such as ATG5, 7, 8 and 12 [50,51] (Figure 2C). In addition, spermidine may be promoted by transcriptional effects on ATGs through the inhibition of Akt and transcriptional activation of FOXO [52], as well as by inhibiting histone acetyl transferases (Figure 2C). In humans, it has been suggested that the levels of spermidine decline with aging, and reduced endogenous concentrations of spermidine may be associated with age-related deterioration of various cellular functions, such as autophagy. The upregulation of endogenous spermidine levels may exert benefits on lifespan extension; however, this effect remains to be examined. Interestingly, Bárcena et al. reported that MetR exhibits lifespan extension and a 10-fold increase in the polyamine spermidine in the livers of Lmna progeroid mice [19], which may indicate that MetR-induced spermidine upregulation is involved in lifespan extension through the induction of autophagy.

### 4.2. Hydrogen Sulphate (H_2_S)

H_2_S has been identified as the third most-prevalent endogenous signaling gas following nitric oxide and carbon monoxide [53]. H_2_S is endogenously generated in organisms, including mammals, through the transsulfuration pathway via two key enzymes, CBS and CGL, as described above. H_2_S can freely diffuse across the cell membrane and has pleiotropic benefits in tissues and organs with the potential to contribute to stress resistance by exerting beneficial effects, including antioxidative effects. The antioxidative effect of H_2_S is exerted through multiple mechanisms. H_2_S acts as a direct scavenger of ROS and upregulates endogenous antioxidant defenses. Kimura et al. demonstrated that H_2_S increases the glutathione levels, which decrease during the cell death cascade, by enhancing GCL activity and upregulating cystine transport, resulting in neuron protection [54]. In addition, Calvert et al. reported that H_2_S shows antioxidative effects through the upregulation of antioxidative molecules, including heme oxygenase-1 and thioredoxin-1, in cardiomyocytes in a NF-E2-related factor 2 (Nrf2)-dependent manner [55].

Dietary restriction of SAAs, including methionine, contributes to stress resistance and lifespan extension through H_2_S production by enhancing the transsulfuration pathway. Previously, Hine et al. reported that the protective effects of 50% DR for 1 week against ischemic-reperfusion injury in the liver are abrogated by the addition of SAAs, including methionine and cysteine, via the inhibition of H_2_S production [56]. In addition, adult mice fed MetR diets for 4 months, and mice with a fasting every other day or 20–30% DR for 6 weeks from 6–8 weeks old led to an increase in H_2_S production capacity in the extracts of the liver and kidney compared to that exhibited in the extracts of control mice fed a complete diet ad libitum (AL) [56]. Furthermore, in fruit flies, the maximal capacity of H_2_S production observed in whole-body extracts of flies fed various levels of DR and MetR was correlated with the maximal lifespan extension [56]. In *C. elegans* and *S. cerevisiae*, the lifespan and/or chronological lifespan was extended in a H_2_S production-dependent manner [56]. The effect of DR on lifespan extension is exerted through increased endogenous H_2_S production via transsulfuration pathway activation upon the restriction of SAAs. On the other hand, the addition of specific SAAs and the activation of mTORC1 suppress the transsulfuration pathway and H_2_S production.

In addition, several reports indicate that MetR protects tissues and organs such as the kidney and heart against metabolic stress and the aging process through increased H_2_S production [57,58]. Wang et al. reported that MetR in mice initiated from 20 months of age slowed kidney senescence and lifespan extension through H_2_S production and AMPK pathway activation [57]. Han et al. also demonstrated that MetR for 15 weeks decreased inflammation and oxidative stress by enhancing the H_2_S production in the heart, and improved cardiac function, in middle-aged (28 weeks old) HFD-induced obese mice [58]. Moreover, MetR for 22 weeks improved hepatic steatosis through increased hepatic H_2_S production in HFD-induced obese mice [59,60]. Interestingly, Xu et al. showed that dietary intervention by MetR for 16 weeks to obese mice fed HFD from 5 weeks of age for 10 weeks can ameliorate impaired learning and memory function by increasing H_2_S production in the hippocampus [61].

### 4.3. Glutathione Synthesis

GSH, a thiol antioxidant, scavenges ROS and is converted to oxidized glutathione (GSSG) in reaction to increased GSH and ROS [62,63]. GSH is produced through the transsulfuration pathway in methionine metabolism as described above. Previous reports showed that 80% MetR increases the level of GSH in erythrocytes in rats [7,64]. Although MetR decreases the GSH levels in several tissues, including the liver [7,64,65,66,67,68,69], oxidative stress is not enhanced. The low levels of hepatic GSH induced by MetR are compensated by increased antioxidative capacity, including increased erythroid 2-like 2 transcription factor (NFE2L2)-induced antioxidative responses [65].

### 4.4. Mitochondrial Oxidative Stress

Several studies have shown that methionine can stimulate mitochondrial ROS (MtROS) production and that MetR inhibits this production. Sanz et al. demonstrated that 80% MetR fed to rats (250 g BW) for 6–7 weeks without CR decreases MtROS production of complexes I and III in the liver and complex I in the heart [70]. Similarly, Caro et al. showed that both 80% and 40% MetR without CR fed for 6–7 weeks to rats (250–300 g BW) decreased the MtROS generation in the liver [71]. The decrease in MtROS was not differentially induced by concentrations of MetR between 40% and 80%, whereas the decrease in mtROS production at MetR concentrations of the control level to 40% was more pronounced, similar to previous observations of the effects of 40% PR and 40% DR [72]. The results indicate that 40% MetR is enough to reduce ROS overproduction in the livers of rats. In addition to the liver, 40% MetR for 7 weeks to rats of 7 weeks of age decreased MitROS production in the brain, kidney and heart [73,74]. These results suggest that the beneficial effect of DR or PR on lifespan extension may be exerted through MetR partially by attenuating MtROS overproduction.

What is the mechanism by which MetR decreases MtROS production? Gomez et al. showed that the direct administration of methionine to isolated functional mitochondria from the liver and kidney of rats increased the rates of mtROS production, suggesting a direct and rapid effect of methionine on mitochondrial complex I [75]. However, there is a possibility that this reaction is mediated by a chemically reactive methionine metabolite. On the basis of electron paramagnetic resonance (EPR) spin trapping and gas chromatography (GC)–flame ionization detector (GC–FID) and GC–mass spectrometry (GC–MS) techniques, it has been reported that the reaction of methionine with hydroxyl radicals produces methionine radical carbon-, nitrogen- and sulfur-centered radicals as intermediates in the formation of the methanethiol product [76]. These radicals or methanethiol (CH3SH) itself may react with complex I or III in mitochondria, resulting in the overproduction of mtROS. Therefore, MetR may decrease the rates of mtROS production through a reduction in the thiolization of complex I in mitochondria.

## 5. Conclusions and Future Prospects

Among dietary interventions, MetR may be a strong candidate for improving longevity and metabolic health, as indicated by the data from animal studies, partially through a reduction in oxidative stress via multiple mechanisms (Figure 3). It is impossible to perform long-term randomized controlled trials to evaluate the effects of dietary interventions, including MetR, on longevity in humans; however, as a healthy dietary pattern, MetR can be considered for maintaining metabolic health, including the prevention of obesity and diabetes. In addition, several pieces of basic and clinical evidence show that MetR has attractive prospects and can effectively inhibit tumor growth as a single or adjuvant agent [77,78,79,80,81,82,83,84,85,86]. However, it remains unclear whether MetR can reduce the risk of cancer occurrence, and the effect of MetR on the suppression of cancer growth is related to a decrease in oxidative stress. Methionine is an essential amino acid; therefore, methionine is required for normal developmental processes, and restriction during embryonic, postnatal or adolescent development can lead to deleterious consequences. On the other hand, the levels of methionine intake that are necessary to benefit health may be age-dependent. Therefore, the level of methionine intake that is required to maintain physiological processes substantially decrease after early adulthood. In many previous clinical studies, a low-methionine diet means a methionine intake of less than 2 mg/kg BW/day, whereas Gao et al. reported that a 3-week MetR diet (2.92 mg/kg/day methionine intake) is sufficient to decrease circulating methionine levels, in healthy and middle-aged individuals [85]. However, the levels of restricted methionine intake that have beneficial effects, including improved longevity and suppression of age-related diseases, such as metabolic disease and cancer, remain unknown in humans. Dietary methionine is contained in animal sources of protein, such as beef, lamb, fish, pork and eggs, at higher levels than plant sources of proteins, which include nuts, seeds, legumes, cereals, vegetables and fruits [87]. Therefore, to achieve MetR, we need to select fewer methionine-rich animal-based foods. Diets that tend to include less methionine, such as vegan, fat-based (e.g., ketogenic) and carbohydrate-based diets (Japanese or DASH: Dietary Approaches to Stop Hypertension), still have protein as the major source of energy [86,87,88,89]. However, animal proteins, which are methionine-rich foods, are an important dietary source of micronutrients, including vitamins and minerals such as iron and zinc; therefore, appropriate intake is necessary to avoid malnutrition.

In addition to diet therapy, adequate exercise is widely considered an important intervention for lifespan extension and promoting healthy aging. Exercise increases ROS production in several tissues, including skeletal muscle or vascular cells [90,91]. Minimum levels of ROS can induce beneficial adaptations by up-regulating cellular antioxidant and oxidative damage repair systems and mitochondrial quality control by autophagy, resulting in suppression of sarcopenia or improvement of endothelial function [91,92,93,94]. In contrast, high-intensity exercise can induce excessive ROS production, and it becomes harmful [90]. Although there may be a difference among individuals on the type and intensity of exercise that can produce the benefits, it is important to select the appropriate intensity, duration, frequency and kind of exercise to exert the beneficial effects of exercise. Thus, a combination of a healthy diet pattern, including MetR and adequate exercise, should be helpful for healthy aging; however, further studies are necessary to clarify the effects and underlying mechanisms.

## Figures and Tables

**Figure 1 biomedicines-09-00130-f001:**
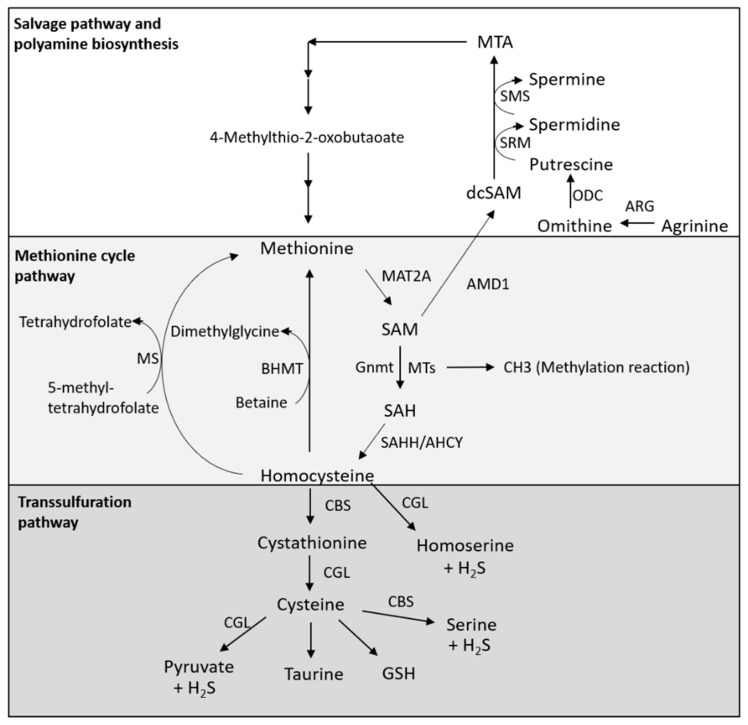
Methionine metabolism. Methionine cycle pathway: Methionine is catabolized by methionine adenosyltransferase 2A (MAT2A), producing the methyl donor S-adenosyl-methionine (SAM). Methyltransferases (MTs), including glycine N-methyltransferase (Gnmt), use SAM as a methyl source, thereby producing S-adenosyl-homocysteine (SAH). SAH is then converted by SAH hydrolase (SAHH, also known as adenosylhomocysteinase, AHCY) to homocysteine. Homocysteine can then either contribute to the transsulfuration pathway for glutathione synthesis or be converted back to methionine by methionine synthase (MS) or betaine homocysteine methyltransferase (BHMT), thus completing the methionine cycle. Salvage pathway and polyamine biosynthesis: Methionine contributes to polyamine biosynthesis by serving as a source of SAM. SAM is decarboxylated by SAM decarboxylase 1 (AMD1) to decarboxylated SAM (dcSAM), which acts as an aminopropyl group donor. Arginine is converted by arginase (ARG) to ornithine and is decarboxylated by ornithine decarboxylase (ODC) to produce putrescine. Putrescine is converted to spermidine and spermine through spermidine synthase (SRM) and spermine synthase (SMS), which both use dcSAM as an aminopropyl donor. On the other hand, dcSAM is converted to MTA after the donation of an aminopropyl group for polyamine synthesis, and MTA is converted by multiple enzymatic steps to methionine. Transsulfuration pathway: Homocysteine produced through the methionine cycle is metabolized in the transsulfuration pathway to generate cysteine. CBS synthesizes cystathionine by the condensation of homocysteine and serine. Thereafter, cystathionine is hydrolyzed by cystathionine-γ-lyase (CGL) to generate cysteine, and cysteine is further utilized in glutathione (GSH) and taurine synthesis. In addition, homocysteine is catalyzed by CGL, producing homoserine and H_2_S, and both CGL and CBS catalyze the H_2_S production from cysteine and the production of pyruvate and serine, respectively.

**Figure 2 biomedicines-09-00130-f002:**
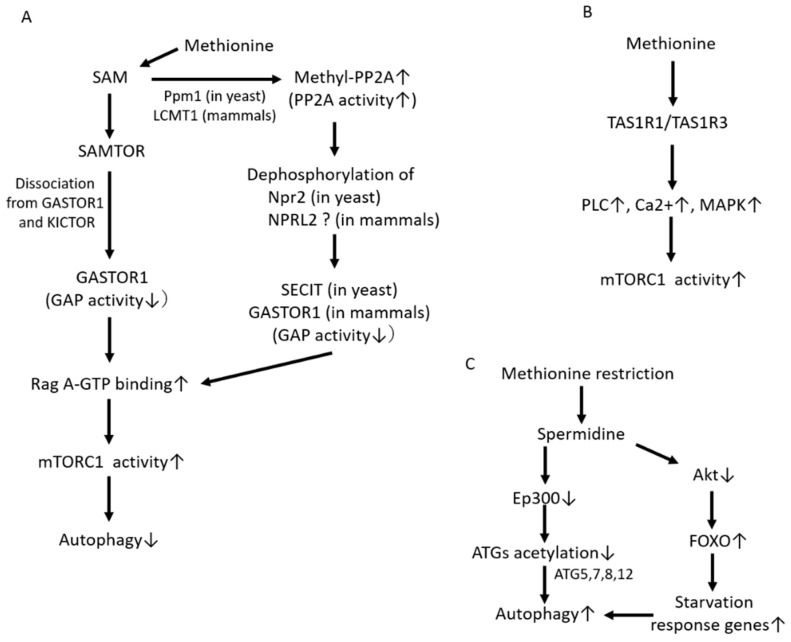
Mechanism of the activation of mTORC1 and the regulation of autophagy by methionine metabolism. (**A**) Regulation of mTORC1 and autophagy by methionine: Intracellular SAM concentration is sensed by SAMTOR, leading to mTORC1 activation and autophagy suppression through the inactivation of GASTOR1, a negative regulator of mTORC1. On the other hand, intracellular SAM promotes the methylation of protein phosphatase 2A (PP2A) via Ppm1 (in yeast) or leucine carboxy methyltransferase 1 (LCMT1) (in mammals), resulting in its activation. Methylated PP2A activates mTORC1 and suppresses autophagy, through the inactivation of SECIT in yeast and GASTOR in mammals via the dephosphorylation of Npr2 in yeast and possibly NPRL2 in mammals. (**B**) TAS1R1–TAS1R3 can serve as a sensor for extracellular methionine, leading to mTORC1 activation through phospholipase C (PLC) activation, an increase in intracellular calcium, and mitogen-activated protein kinase (MAPK) activation. (**C**) Spermidine induced by methionine restriction can promote autophagy by decreasing the acetylation of autophagy-related genes (ATGs) by suppressing Ep300. In addition, spermidine can promote autophagy through the upregulation of starvation response gene expression induced by the suppression of Akt and the transcriptional activation of FOXO. ↑: upregulation, ↓: downregulation.

**Figure 3 biomedicines-09-00130-f003:**
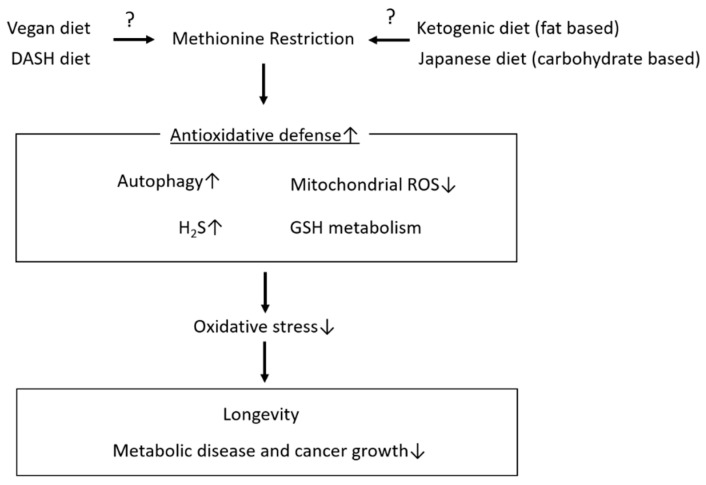
Effect of methionine restriction on longevity and age-related pathologies. Methionine restriction may be one of the dietary interventions involved in longevity and the amelioration of age-related pathologies, including metabolic disease and cancer growth, partially through the enhancement of antioxidative defenses and attenuation of oxidative stress. Several diet patterns, such as the vegan diet, the DASH diet, ketogenic diet (fat based) and Japanese diet (carbohydrate based), may be candidates for a dietary pattern that restricts methionine intake. ↑: upregulation, ↓: downregulation.

## Data Availability

Not applicable.

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
