# Peer review of "Effect of Methionine Restriction on Aging: Its Relationship to Oxidative Stress"

_biomedicines, 2021, doi:10.3390/biomedicines9020130_

Round 1

Reviewer 1 Report

The authors review the topic of methionine restriction (MetR) in an aging population, to potentially ameliorate age-associated oxidative stress.  In addition to reviewing MetR's impact on both animal and human models, they also review several proposed molecular mechanisms whereby MetR provides a beneficial effect.

General comments:

1) In section 3, the authors discuss the impact of MetR in animal studies more so than human studies.  The reviewer acknowledges that much more work has been done investigating the effect of MetR in animals and lower organisms compared to humans, but can they provide additional newer references for the human studies (i.e. have they included recent calorie restriction publications that effectively reduce methionine consumption?).  

2) Have the proposed mechanisms in Section 4 all been observed in aging subjects?  This should be clarified in each sub-section.

3) Can the authors comment about the impact of exercise on oxidative stress, given that healthy aging is commonly associated with both dietary restriction and increased physical activity?

Author Response

Response to reviewers

----- Here we sincerely appreciated editor’s suggestions of our manuscript. We did response all the concerns from all reviewers. Reviewers’ comments were helpful and also making our manuscript quality better one.

Reviewer(s') comments:

The authors review the topic of methionine restriction (MetR) in an aging population, to potentially ameliorate age-associated oxidative stress.  In addition to reviewing MetR's impact on both animal and human models, they also review several proposed molecular mechanisms whereby MetR provides a beneficial effect.

General comments:

1) In section 3, the authors discuss the impact of MetR in animal studies more so than human studies. The reviewer acknowledges that much more work has been done investigating the effect of MetR in animals and lower organisms compared to humans, but can they provide additional newer references for the human studies (i.e. have they included recent calorie restriction publications that effectively reduce methionine consumption?). 

Ans)

We appreciate for the reviewer’s comments. We searched again for research papers that examined the effects of methionine restriction on anti-aging or metabolic health, in humans. However, we couldn't find any other new papers. A calorie-restricted diet usually limits 30-40% of all nutrients, whereas a methionine-restricted diet needs to be limited by 80% for effectiveness on lifespan extension or improving metabolic health. Therefore, calorie-restricted diet can not achieve sufficient level of methionine restriction. Dietary patterns that can achieve a low methionine diet include a vegan diet, a fat-based ketogenic diet with low in animal protein, a carbohydrate-based diet (Japanese diet or DASH diet), as mentioned in the section of conclusions. However, there are currently no reports of human studies examining the effects of these dietary patterns on anti-aging and metabolic health from the perspective of methionine restriction.

2) Have the proposed mechanisms in Section 4 all been observed in aging subjects?  This should be clarified in each sub-section.

Ans)

We appreciate for the reviewer’s comments. To clarify whether the proposed mechanisms in Section 4 have been observed in aging animal models, we changed several sentences, as below.

  • In addition, adult mice fed MetR diets for 4 months, and mice with a fasting every other day or 20–30% DR for 6 weeks from 6-8 weeks old led to an increase in H2S production capacity in the extracts of the liver and kidney compared to that exhibited in the extracts of control mice fed a complete diet ad libitum (AL). (Section. 4.2.)
  • Wang et al. reported that MetR in mice initiated from 20 months of age slowed kidney senescence and lifespan extension through H2S production and AMPK pathway activation. Han et al. also demonstrated that MetR for 15 weeks decreased inflammation and oxidative stress by enhancing H2S production in the heart, and improved cardiac function, in middle-aged (28 weeks old) HFD-induced obese mice. Moreover, MetR for 22 weeks improved hepatic steatosis through increased hepatic H2S productionin HFD-induced obese mice. Interestingly, Xu et al. showed that dietary intervention by MetR for 16 weeks to obese mice fed HFD from 5 weeks of age for 10 weeks can ameliorate impaired learning and memory function by increasing H2S production in the hippocampus. (Section. 4.2.)
  • Sanz et al. demonstrated that 80% MetR fed to rats (250g BW) for 6-7 weeks without CR decreases MtROS production of complexes I and III in the liver and complex I in the heart. Similarly, Caro et al. showed that both 80% and 40% MetR without CR fed for 6-7 weeks to rats (250-300g BW) decreased MtROS generation in the liver. (Section. 4.4.)
  • In addition to the liver, 40% MetR for 7 weeks to rats with 7 weeks of age decreased MitROS production in the brain, kidney and heart. (Section. 4.4.)

3) Can the authors comment about the impact of exercise on oxidative stress, given that healthy aging is commonly associated with both dietary restriction and increased physical activity?

Ans)

We appreciate for the reviewer’s comments. This review paper focuses on MetR diet and aging. Therefore, we added the discussion regarding the relationship between exercise and oxidative stress on aging in the section of conclusions and future prospects, as below.

“In addition to diet therapy, adequate exercise is widely considered an important intervention for lifespan extension and promoting healthy aging. Exercise increases ROS production in several tissues including skeletal muscle or vascular cells. Minimum levels of ROS can induce beneficial adaptations by up-regulating cellular antioxidant and oxidative damage repair systems, and mitochondrial quality control by autophagy, resulting in suppression of sarcopenia or improvement of endothelial function. In contrast, high-intensity exercise can induce excessive ROS production, and it becomes harmful. Although there may be a difference among individuals on the type and intensity of exercise that can produce the positive effects, it is important to select the appropriate intensity, duration, frequency, and kind of exercise to be exerted the beneficial effects of exercise. Thus, combination of healthy diet pattern, including MetR and adequate exercise should be helpful for health aging; however, further studies are necessary to clarify the effects and underling mechanisms.”

Reviewer 2 Report

The manuscript, entitled "Effect of methionine restriction on aging: its relationship to oxidative stress", summarizes the current state of research on the mechanisms underlying lifespan extension by methionine restriction, particularly as they apply to oxidative damage and autophagy.  The review is thorough and fairly comprehensive, and given the potential of this intervention to improve healthspan in humans, this work represents an important resource to the aging and biogerontology fields.  However, there are several issues that need to be resolved by the authors before the manuscript is suitable for publication.

1. Although the manuscript is well-written in parts, there are pervasive English language issues that make other parts essentially unreadable.  As a result, the manuscript requires extensive English language editing.  While by no means a comprehensive list, several instances of incorrect grammar or otherwise poor English are indicated below in order to demonstrate the issue.

A. Lines 45-47.  "In addition, glucose restriction, in yeast down-regulates the
transcription and translation of methionine biosynthetic enzymes and transporters, leading to a decreased intracellular methionine concentration, thereby may extend its lifespan ..."

B. Line 50.  "... oxidative stress via multiple mechanisms, as described as follows ..."

C. Lines 61-62.  "Methionine ... is supplied by autophagy ..."  This sentence, as written, implies that autophagy is the sole source of methionine, which is problematic, particularly as the preceding sentence indicates that methionine is provided by the diet.

D. Lines 138-139.  "Metabolically, reducing adiposity in rodents, while MetR leads to a paradoxical increase in both energy intake and energy expenditure ..."

2. In Figure 1, the diagram needs to be altered so that secondary reactants in enzymatic reactions (e.g., putrescine to spermidine) are connected by curved arrows, not straight ones (see metabolic pathway diagrams in other articles for examples).  As composed, the diagram is highly confusing.

3. In section 3.2, dealing with results from human studies, the work of Johnson at al. (2014; Methionine restriction activates the retrograde response and confers both stress tolerance and lifespan extension to yeast, mouse and human cells) and Gao et al. (2019; Dietary methionine influences therapy in mouse cancer models and alters human metabolism) pertaining to cultured human cells and human dietary studies, respectively, needs to be described and cited.

4. The manuscript attempts to make the argument that methionine restriction improves healthspan by activating autophagy to combat the production of ROS caused by dysfunctional mitochondria.  As a result, it is imperative that mitophagy (i.e., the selective degradation of damaged and/or dysfunctional mitochondria by autophagy) is explicitly defined.  Furthermore, it is necessary to describe the findings of Plummer et al. (2019; Extension of cellular lifespan by methionine restriction involves alterations in central carbon metabolism and Is mitophagy-dependent) that implicate mitophagy in the benefits of this intervention to yeast.

Author Response

Response to reviewers

----- Here we sincerely appreciated editor’s suggestions of our manuscript. We did response all the concerns from all reviewers. Reviewers’ comments were helpful and also making our manuscript quality better one.

The manuscript, entitled "Effect of methionine restriction on aging: its relationship to oxidative stress", summarizes the current state of research on the mechanisms underlying lifespan extension by methionine restriction, particularly as they apply to oxidative damage and autophagy.  The review is thorough and fairly comprehensive, and given the potential of this intervention to improve healthspan in humans, this work represents an important resource to the aging and biogerontology fields.  However, there are several issues that need to be resolved by the authors before the manuscript is suitable for publication.

  1. Although the manuscript is well-written in parts, there are pervasive English language issues that make other parts essentially unreadable. As a result, the manuscript requires extensive English language editing.  While by no means a comprehensive list, several instances of incorrect grammar or otherwise poor English are indicated below in order to demonstrate the issue.

Ans)

Our manuscript was edited by Springer Nature Author Service.

  1. Lines 45-47. "In addition, glucose restriction, in yeast down-regulates the

transcription and translation of methionine biosynthetic enzymes and transporters, leading to a decreased intracellular methionine concentration, thereby may extend its lifespan ..."

Ans)

In addition, in yeast, glucose restriction down-regulates the transcription and translation of methionine biosynthetic enzymes and transporters, leading to a decreased intracellular methionine concentration and lifespan extension.

  1. Line 50. "... oxidative stress via multiple mechanisms, as described as follows ..."

Ans)

Oxidative stress is closely related to age and impaired metabolic health. Therefore, MetR-induced beneficial effects on lifespan extension and metabolic health are mediated partially through a reduction in oxidative stress via multiple mechanisms, as described as follows:

  1. Lines 61-62. "Methionine ... is supplied by autophagy ..." This sentence, as written, implies that autophagy is the sole source of methionine, which is problematic, particularly as the preceding sentence indicates that methionine is provided by the diet.

Ans)

Methionine is an essential amino acid that is necessary for normal growth and development, and functions as an initiator of protein synthesis. Methionine is also a sulfur‐containing amino acid (SAA). In humans, methionine is obtained from both food and gut microbes, and is also supplied by autophagy, as described herein.

  1. Lines 138-139. "Metabolically, reducing adiposity in rodents, while MetR leads to a paradoxical increase in both energy intake and energy expenditure ..."

Ans)

Metabolically, MetR reduces adiposity in rodents, whereas interestingly, both energy intake and energy expenditure (EE) are increased.

  1. In Figure 1, the diagram needs to be altered so that secondary reactants in enzymatic reactions (e.g., putrescine to spermidine) are connected by curved arrows, not straight ones (see metabolic pathway diagrams in other articles for examples). As composed, the diagram is highly confusing.

Ans)

We appreciate for the reviewer’s comments. We improved Figure 1, as the reviewer pointed out.

  1. In section 3.2, dealing with results from human studies, the work of Johnson at al. (2014; Methionine restriction activates the retrograde response and confers both stress tolerance and lifespan extension to yeast, mouse and human cells) and Gao et al. (2019; Dietary methionine influences therapy in mouse cancer models and alters human metabolism) pertaining to cultured human cells and human dietary studies, respectively, needs to be described and cited.

Ans)

We appreciate for the reviewer’s comments. We cited the paper written by Johnson et al. 2014, and inserted the sentence “In addition to MetR, genetic Meth-R by knockdown of methionine synthase confers stress resistance in cultured fibroblasts, as well as a reduced doubling time and a replicative lifespan extension.” in the section of 3. Effects of MetR on life span extension and metabolic health.

Our current review paper focus on role of MetR diet on aging and metabolic health. Previous research paper written by Gao et al. 2019 described the effect of MetR diet on cancer therapy. Therefore, we cited Gao’s paper just in the section of conclusions and future prospects in a first submitted manuscript, as below. “In addition, several pieces of basic and clinical evidence show that MetR has attractive prospects and can effectively inhibit tumor growth as a single or adjuvant agent.” In addition, we added the sentence; “Gao et al. reported that a 3-week of MetR diet (2.92mg/kg/day methionine intake) is sufficient to decrease circulating methionine levels, in healthy and middle-aged individuals”, in the section of conclusions and future prospects.

  1. The manuscript attempts to make the argument that methionine restriction improves healthspan by activating autophagy to combat the production of ROS caused by dysfunctional mitochondria. As a result, it is imperative that mitophagy (i.e., the selective degradation of damaged and/or dysfunctional mitochondria by autophagy) is explicitly defined. Furthermore, it is necessary to describe the findings of Plummer et al. (2019; Extension of cellular lifespan by methionine restriction involves alterations in central carbon metabolism and Is mitophagy-dependent) that implicate mitophagy in the benefits of this intervention to yeast.

Ans)

We appreciate for the reviewer’s comments. We cited the paper written by Plummer et al. 2019, and inserted the sentences, “Among the autophagy, damaged mitochondria are eliminated by selective autophagy, mitophagy, leading to the induction of mitochondrial biogenesis and suppression of oxidative stress. Indeed, Plummer et al. reported that MetR-mediated chronological lifespan extension of the yeast requires the autophagic recycling of mitochondria, that is mitophagy.” in the section of 4.1 induction of Autophagy.